# Structural characterization of the equatorial F region plasma irregularities in the multifractal context

Neelakshi J[1], Reinaldo R. Rosa[1], Siomel Savio[2,3], Esfhan Alam Kherani[2], Francisco Carlos de Meneses[4], Stephan Stephany[1], and Polinaya Muralikrishna[2]

[1]Computational Space Physics Group, Lab for Computing and Applied Math (LABAC), National Institute for Space Research (INPE), Av. dos Astronautas, 1758, São José dos Campos, São Paulo 12227-690, Brazil

[2]Aeronomy Division, National Institute for Space Research (INPE), Av. dos Astronautas, 1758, São José dos Campos, São Paulo 12227-690, Brazil

[3]China-Brazil Joint Laboratory for Space Weather, NSSC/INPE, Av. dos Astronautas, 1758, São José dos Campos, São Paulo 12227-690, Brazil

[4]School of Physics and Mathematics, Autonomous University of Nuevo León (UANL), Av. Universidad s/n, Cd. Universitaria, San Nicolás de los Garza, N.L. 66455, Mexico

**Correspondence:** Neelakshi J. (neelakshij@gmail.com)

**Abstract.** In the emerging ionosphere-space-weather paradigm, investigating the dynamical properties of ionospheric plasma irregularities using advanced computational nonlinear algorithms is providing new insights into their turbulent-like nature, for instance, the evidence of energy distribution via a multiplicative cascade. In this study, we present a multifractal analysis of the equatorial F region *in situ* data obtained from two different experiments performed at Alcântara (2.4°S; 44.4°W), Brazil to explore their scaling structures. The first experiment observed several large-medium scale plasma bubbles whereas the second experiment observed vertical uplift of the base of F region. The multifractal detrended fluctuation analysis and the p-model fit are used to analyse the plasma density fluctuation time series. The result shows the presence of multifractality with degree of multifractality $0.53 - 0.93$ with $0.3 \leq p \leq 0.4$ cascading probability for the first experiment. Another experimental data also exhibits multifractality with degree of multifractality $0.19 - 0.27$ with $0.42 \leq p \leq 0.44$ cascading probability in ionospheric plasma irregularities. Our results confirm the nonhomogeneous nature of plasma irregularities and characterize the underlying nonhomogeneous multiplicative cascade hypothesis in the ionospheric medium. Differences in terms of scaling and complexity in the data belonging to different types of phenomena are also addressed.

## 1  Introduction

Present ionospheric research is transiting towards ionospheric space weather that goes beyond the ground- and space-based communication interruptions to influence decision making communities on social, economical, and physical infrastructural policies. The enhancements in ionospheric plasma irregularities driven by space weather conditions demand an accurate characterization of the dynamical properties of the electron density and its complex nonlinear variation (Cander, 2019). With instruments operating over a substantial frequency domain, a study of plasma density irregularities provide insight into the underlying physical mechanism and its structural properties (Wernik et al., 2003; Muralikrishna et al., 2003). Energy dissipation

is found to be an underlying process for the occurrence of electron density or electric field fluctuations in ionospheric plasma irregularities (Jahn and LaBelle, 1998; Kelley and Hysell, 1991). Various rocket experiments and numerical simulations have been performed and contributed to our understanding of the generation and development of ionospheric irregularities. Costa and Kelley (1978) showed that the Rayleigh-Taylor instability that initiates in the bottomside equatorial F-region can non-linearly develop very sharp gradients leading to the formation of steepened structures responsible for the power-law spectra observed by a rocket experiment in Natal, Brazil. Shock waves were observed by numerical simulation performed by Zargham and Seyler (1987) of the generalized Rayleigh Taylor instability at the bottomside and topside F-region equatorial ionosphere, which was confirmed by rocket and satellite *in situ* data reported by Kelley and Seyler and Zargham (1987). Hysell et al. (1994a, b) proposed a model of plasma steepening, evolving from plasma advection that occurs on the vertical leading edges of plasma depletion wedges, to interpret shock waves detected in the equatorial ionosphere by rockets launched from Kwajalein Atoll. Jahn and LaBelle (1998) measured shocklike structures characterized by the density waveforms at the bottomside and topside F-region of the equatorial ionosphere in a rocket experiment in Alcântara, Brazil.

The spectral analysis though widely used, falls short in characterizing nonstationary data as they assume stationarity in the data, which is equivalent to presuming homogeneous turbulence; hence, a more robust method is necessary to analyse nonstationary data (Wernik et al., 2003). In addition, to develop a robust specification and a forecasting model, along with classical morphological, statistical and spectral studies, a thorough understanding of nonlinearity in ionospheric irregularities is essential (Tanna and Pathak, 2014).

Recent advances in the computational algorithms based on fractal formalism, supplemented with mathematical modelling derived from probabilistic measures have conclusively substantiated the occurrence of the energy cascading process in turbulent sites in the solar and interplanetary environment as well as in the laboratory using Kolmogorov's formalism as the basis (Grauer et al., 1994; Carbone et al., 1995; Abramenko et al., 2002; Macek, 2007; Wawrzaszek and Macek, 2010; Chian and Muñoz, 2011; Miranda et al., 2013; Wawrzaszek et al., 2019).

Various different approaches had been explored to understand nonlinear characteristics and intermittency in ionospheric irregularities, like structure function analysis (Dyrud et al., 2008; Spicher et al., 2015), fractal and multifractal analysis (Wernik et al., 2003; Alimov et al., 2008; Bolzan et al., 2013; Tanna and Pathak, 2014; Miriyala et al., 2015; Chandrasekhar et al., 2016; Fornari et al., 2016; Sivavaraprasad et al., 2018; Neelakshi et al., 2019), and multispectral optical imaging (Chian et al., 2018).

Structure function analysis performed on ionospheric high latitude *in situ* data have revealed the intermittent nature of ionospheric irregularities owing to the large deviations from the Kolmogorov's K41 universal power-law index proposed for neutral fluid turbulence (Spicher et al., 2015).

In all the above mentioned studies, the main feature which gets highlighted is that the power spectra point to large deviations from the homogeneous turbulence described by the Kolmogorov spectrum ($-5/3$). Also, higher order statistics like structure function analysis confirmed the deviation from the Kolmogorov scales. Thus affirming the nonhomogeneity and intermittency in ionospheric irregularities. In the complex scenario of ionospheric turbulence, an important question that arises in the context of this paper is, "is nonhomogeneity, which can be characterized by multifractal spectra, the cause for the large deviations

from the $-5/3$?" To answer this question, we propose to use the multifractal detrended fluctuation analysis (MFDFA) on the equatorial F region plasma irregularities.

A detrended fluctuation analysis (DFA; Peng et al. (1994)) has been a proven successful method to find a power law correlation and monofractal scaling in noisy, nonstationary data. The DFA is a robust method as it can handle discontinuous and length-wise short data. In case data is more complex and has intricate scaling, various scaling exponents characterize different parts of the data. To characterize such multiple scaling behavior in the data, Kantelhardt et al. (2002) generalized DFA to a multifractal detrended fluctuation analysis (MFDFA), and have shown the equivalence to standard partition function based multifractal method for stationary data with compact-support.

The MFDFA has wide applications in many branches of sciences, such as medicine (Makowiec, 2011), physics (Freitas et al., 2016), engineering (Lu et al., 2016), finance (Grech, 2016), and social sciences (Kantelhardt, 2009; Telesca and Lovallo, 2011), to understand the complexity of a system through its scaling exponents that characterize multifractal dynamics of the system. The MFDFA has been applied to study ionospheric scintillation index time series (Tanna and Pathak, 2014; Miriyala et al., 2015) and ionospheric total electron content data (Chandrasekhar et al., 2016; Sivavaraprasad et al., 2018). In analogy, a wavelet transform is applied to study ionospheric irregularities (Wernik et al., 2003; Bolzan et al., 2013). These analyses identified multifractality and intermittency in nonlinear ionospheric irregularities.

In this work, we explore the low latitude equatorial F region *in situ* data obtained from two different experiments and performed from the same rocket launching station. In the first experiment, done on 18 December 1995, the rocket traversed through various large-medium scale plasma irregularities during its descent, which were associated with the generalized Rayleigh-Taylor instability (Muralikrishna et al., 2003). Whereas in the second experiment, done on 8 December 2012, the base of the F region was moving upward, i.e., pre-reversal enhancement (PRE) of vertical plasma drift was observed (Savio et al., 2016; Savio Odriozola et al., 2017).

In the equatorial ionosphere, the evening PRE is considered as an important seeding mechanism for the post sunset F region irregularities, as quick and acute uplift of the electric field escalates the rate of growth of the generalized Rayleigh-Taylor instability (Li et al., 2007; Kelley et al., 2009; Abdu et al., 2018). Knowing the relation between these two phenomena, it will be interesting to know the differences in their scaling behavior and complexity. Investigating these plasma fluctuations may enable to study the scaling properties of these plasma irregularities, and also knowing various characteristics along with the complexity of data may provide important inputs to model empirical data. Hence, we apply the MFDFA method to the plasma density fluctuation data obtained from these two different *in situ* experiments. To corroborate our results, a multifractal spectrum obtained from the MFDFA is fitted with the p-model (Meneveau and Sreenivasan, 1987) based on the generalized two-scale Cantor set. Details on the experiments are given briefly in section 2. Methods are described in section 3. The results of the analyses are discussed in section 4 followed by concluding remarks in section 5.

## 2 *in situ* experiments

The equatorial launching station of Brazil is located at Alcântara (2.24° S, 44.4° W, dip latitude 5.5°S). The SONDA III rocket was launched at 21:17 LT, on December 18, 1995 under favourable conditions for formation of a plasma bubble. During the $\sim$ 11 min flight, plane of rocket trajectory was almost orthogonal to the geomagnetic field lines, spanned $\sim$ 589 km distance horizontally with an apogee at altitude $\sim$ 557 km. A rocket-born electric field double probe (EFP) measured electric field fluctuations related to ionospheric plasma irregularities. In the upleg profile (ascent of the rocket), the F region base is clearly observed around 300 km, but without any large scale depletion or bubble. On the other hand, several plasma bubbles of medium-large scale were observed in the downleg profile (descent of the rocket), around the base of F region and also topside of it, but without any sharp indication of the F region base from an altitude above 240 km. The rocket traversed through regions of different altitudes separated by a few hundreds of kilometers during upleg and downleg so this might elucidate the large differences observed in ascent and descent of the rocket (Muralikrishna et al., 2003; Muralikrishna and Abdu, 2006; Muralikrishna and Vieira, 2007). Detail explanation of *in situ* experiment and the analysis is found in Muralikrishna et al. (2003); Muralikrishna and Abdu (2006); Muralikrishna and Vieira (2007).

Some of the key results from the aforementioned (Muralikrishna et al., 2003; Muralikrishna and Abdu, 2006; Muralikrishna and Vieira, 2007) analysis indicate - (1) initiation of cascade process, owing to the generalized Rayleigh–Taylor instability mechanism, near the base of F region that resulted in the development of plasma bubbles or large scale irregularities, and (2) subsequently, advecting energy to higher altitudes, smaller scale irregularities were observed, owing to the Cross-Field instability mechanism.

From the same rocket launching station, Alcântara, a two-stage VS-30 Orion sounding rocket was launched at 19:00 LT, on December 8, 2012, under favorable conditions for strong spread-F. During the $\sim$ 11 min flight, the rocket trajectory was in the north-northeast direction towards the magnetic equator, ranging $\sim$ 384 km horizontally with an apogee at $\sim$ 428 km. A conical Langmuir probe onboard the rocket measured the electron density fluctuations associated with ionospheric plasma irregularities. In this experiment, the F region base was clearly observed in the downleg profile around 300 km, with some small scale fluctuations in the F region. At the rocket launch time, ground equipment, digisonde, operated from equatorial station reported fast uplift of the base of F layer, thus indicating the pre-reversal enhancement of the F region vertical drift (Savio et al., 2016; Savio Odriozola et al., 2017). Further explanation of *in situ* experiment and data analysis is found in Savio et al. (2016); Savio Odriozola et al. (2017).

## 3 Methods

### 3.1 Multifractal detrended fluctuation analysis

Multifractal detrended fluctuation analysis (Kantelhardt et al., 2002) has applied to investigate the multifractal properties of ionospheric irregularities in the following way.

To implement the MFDFA, a plasma density time series $x(k)$ of length $k = 1, 2, ...., N$ is considered. A first step is to compute profile, $Y(i)$, by calculating cumulative sum by subtracting its mean.

$$Y(i) = \sum_{k=1}^{i} [x(k) - \langle x \rangle], \qquad i = 1, ..., N \tag{1}$$

Divide the integrated profile into non-overlapping and equidistant $N_s$ segments of $s$ elements, referred to as scales. The length of the series may not be a multiple of all scales and small part of the profile may be left out. To avoid it, repeat the same procedure over profile but starting from endpoint, in reverse direction.

Now we have total of $2N_s$ segments. These segments are then detrended using linear least squares. The variance is calculated over all segments.

$$F^2(s, v) = \left( \frac{1}{s} \sum_{i=1}^{s} [Y[(v-1)s + i] - y_v(i)]^2 \right) \qquad for\ each\ segment\ v,\ v = 1, 2, ..., N_s \tag{2}$$

and

$$F^2(s, v) = \left( \frac{1}{s} \sum_{i=1}^{s} [Y[N - (v - N_s)s + i] - y_v(i)]^2 \right) \qquad for\ each\ segment\ v,\ v = N_s + 1, ..., 2N_s \tag{3}$$

$y_v(i)$ is a polynomial fit obtained on a segment $v$. Now, averaging over all segments, $q^{th}$ order fluctuation function is computed.

$$F_q(s) = \left( \frac{1}{2N_s} \sum_{v=1}^{2N_s} [F^2(s, v)]^{q/2} \right)^{1/q} \qquad for\ q \neq 0 \tag{4}$$

When $q = 0$, logarithmic averaging should be used to calculate fluctuation function.

$$F_0(s) = exp \left( \frac{1}{2N_s} \sum_{v=1}^{2N_s} [ln(F^2(s, v))] \right) \tag{5}$$

Applying a linear fit to the fluctuation function profile on the log-log plot yields the generalized Hurst exponent, $h(q)$, for each moment $q$ as $F_q(s) \propto s^{h(q)}$. The computed generalized Hurst exponent $h(q)$ can be related to the classical multifractal scaling (or mass) exponent as $\tau(q)$ by $\tau(q) = qh(q) - 1$. The multifractal spectrum is calculated using $h(q)$ as:

$$\alpha = h(q) + qh'(q) \qquad where \quad h'(q) = \frac{dh}{dq} \tag{6}$$

$$f(\alpha) = q(\alpha - h(q)) + 1 \tag{7}$$

where $\alpha$ represents the multifractal strength and $f(\alpha)$ represents a set of multifractal dimensions.

## 3.2 p-model

The p-model is proposed by Meneveau and Sreenivasan (1987) to model the energy cascading process in the inertial range of fully developed turbulence for the dissipation field. The p-model starts with a coherent structure with an assumed specific energy flux per unit length which then undergoes a binary fragmentation at each cascading step, distributing the energy flux with probabilities $p_1$ and $p_2$ among the fragments $l_1$ and $l_2$. In this cascading process, $n$ denotes the number of generations. In each generation, the segment size is given by $l_1^m l_2^{n-m}$ where $m$ denotes the number of left side fragments and $n-m$ represents right side fragments in a segment (Halsey et al., 1986). An analytical formulation for the generalized two scale Cantor set is given by

$$\alpha = \frac{ln(p_1) + (n/m - 1)ln(p_2)}{ln(l_1) + (n/m - 1)ln(l_2)} \tag{8}$$

$$f(\alpha) = \frac{(n/m - 1)ln(n/m - 1) - (n/m)ln(n/m)}{ln(l_1) + (n/m - 1)ln(l_2)} \tag{9}$$

is useful to determine the generalized multifractal dimensions which represent the multifractal spectrum. (Halsey et al., 1986).

Based on the generalized two-scale Cantor set, the p-model consider equal fragment length ($l_1 = l_2$) and unequal weights ($p_1 \neq p_2$ and $p_1 + p_2 \leq 1$). When $p_1 + p_2 \leq 1$, loss in $p$ parameter given by $dp = 1 - p_1 - p_2$, accounts for the direct energy dissipation in the energy cascading process in the inertial range. The proposed p-model claims to display all multifractal properties of one-dimensional section of the dissipation field for fully developed turbulence. The multifractality ceases to exist for $p = 0.5$.

## 4 Results and Interpretation

Six time series of *in situ* observations of electric field fluctuations from the F region are selected from the first experiment performed on 18 December 1995, corresponding to the mean heights of $264.58, 270.22, 292.37, 324.00, 358.56$, and $429.65$ km in downleg. Similarly, from the second experiment performed on 12 December 2012, we selected three time series of electron density fluctuations from the F region, corresponding to the mean heights of $339.94, 348.99$, and $400.24$ km in downleg. These time series are subjected to the multifractal analysis. Primarily, the profile is obtained by differencing the time series i.e. $y = x(i+1) - x(i)$, using the criterion based on the power exponent obtained in the DFA method, prescribed by Ihlen (2012) in Table 2, for biomedical time series, to yield the best results from the MFDFA method. We found the criterion to hold for ionospheric *in situ* data under study. Scales upto one-tenth of the length of the time series are considered. From the MFDFA, the generalized Hurst exponent, $h(q)$, classical multifractal scaling exponent $\tau(q)$ and multifractal spectrum $\alpha$ and $f(\alpha)$ are obtained. We show a comprehensive analysis for only one time series from each of the two experiments (Figures 1 and 2). For the remaining time series, we show only the multifractal spectrum along with its respective time series (Figures 3 and 4), but we report the analysis of both experiments in Tables 1 and 2 respectively.

In the MFDFA, fluctuation function $Fq(s)$ is obtained by computing $q^{th}$ order local root mean square (RMS) for multiple segment size, i.e., for scales $s$. A segment may contain smaller to larger fluctuations. Rapid variation in fluctuations influence overall RMS for smaller scale sizes whereas slow variation in fluctuations influence overall RMS for larger scale sizes. Negative $q$ values characterize smaller fluctuations and positive $q$ values characterize larger fluctuations in a segment. When $q = 0$, it behaves neutral. $h(q)$ has dependence on $q$. To outline, for a multifractal time series $h(q)$ monotonically decreases with $q$, and $\tau(q)$ shows nonlinear dependence on $q$. With $q = 0$ as a center point, let us inspect how $h(q)$ varies with respect to negative and positive values of $q$. If time series is influenced by smaller fluctuations, then variation of $h(q)$ for negative $q$ will be faster, i.e., a steeper slope can be observed with respect to negative $q$ and vice-versa (Kantelhardt et al., 2002; Ihlen, 2012). The multifractal spectrum illustrates how segments with small and large fluctuations deviate from the average fractal structure. The shape and width of multifractal spectrum are also important measures to quantify the nature of multifractality present in the data. For $f(\alpha) = 1$, the corresponding value of $\alpha$, known as $\alpha_0$ divides the spectrum into left and right sides. A shape of the spectrum, difference in left and right side of the spectrum, can be quantified by measure of asymmetry, $A$, is given by

$$A = \frac{\alpha_0 - \alpha_{min}}{\alpha_{max} - \alpha_0} \tag{10}$$

When $A = 1$, the multifractal spectrum is symmetric in the sense that the time series is influenced by both larger as well as smaller fluctuations. When $A > 1$, the spectrum is left-skewed which implies that the time series is more influenced by the larger fluctuations. When $A < 1$, the spectrum is right-skewed which implies that the time series is more influenced with smaller fluctuations.

A width of the spectrum can be quantified by $\Delta\alpha$, which is the difference between maximum and minimum dimension.

$$\Delta\alpha = \alpha_{max} - \alpha_{min} \tag{11}$$

The width of the spectrum infers the degree of multifractality and complexity of the data. It represents the deviation from the average fractal structure, and directly relates to the parameters corresponding to the multiplicative cascade process. Larger (smaller) value of $\Delta\alpha$ infers stronger (weaker) multifractality in the data.

The multifractal spectrum reflects the characteristics of the $h(q)$ profile. In the spectrum, contrary to the $h(q)$ profile, left side is characterized by positive values of $q$, and right side is characterized by negative values of $q$. When the $h(q)$ profile show steeper variations on left side, i.e., for negative $q$'s, right side of the spectrum shows faster variation compared to its left side.

Figure 1 shows a detailed multifractal analysis of a time series from the first experiment, corresponding to the mean height of 324.00 km (top left). The profile of $h(q)$ as a function of $q$ is shown on the top right, and of $\tau(q)$ on the bottom left. The corresponding multifractal spectrum is shown on the bottom right panel. The spectrum is right-skewed, indicating the influence of the negative values of $q$ on the data. It is evident as well from the $h(q)$ profile as the variation of $h(q)$ for negative $q$ is observed to be comparatively steep. The plot for $\tau(q)$ versus $q$ shows marked deviation from the linearity, asserting the presence of the multifractality in time series for the chosen height. In addition to the derived inferences from the visual analysis of the multifractal spectrum reported above, multifractal measures, $\Delta\alpha$ and $A$ can be quantified (Eqs. 11 and 10). Measure $A = 0.32$ quantifies the skewness while $\Delta\alpha = 0.72$ infers the strength of multifractality. These two measures are listed in Table

1. Lastly, the multifractal spectrum is fitted with the p-model (shown with a continuous line), where the fragment lengths are equal i.e., $l_1 = l_2 = 0.5$ and the weights, $p_1$ and $p_2$, are varied such that $p_1 + p_2 \leq 1$. Nevertheless, loss in $p$ parameter had to be accounted to obtain an optimal fit. The loss factor, $dp$, signifies nonconservative energy distribution i.e., a dissipative energy cascading process in the inertial range. We have obtained a dissipative factor of 0.090, with $p_1 = 0.315$. The p-model fit parameters are listed in Table 1.

Similar to Figure 1, Figure 2 shows a detailed multifractal analysis of a time series from the second experiment, corresponding to the mean height of 339.94 km (top left). The profile of $h(q)$ as a function of $q$ is shown on the top right, and of $\tau(q)$ on the bottom left. The corresponding multifractal spectrum is shown on the bottom right panel. The spectrum is left-skewed, indicating the influence of the positive values of $q$ on the data. The variation of $h(q)$ for positive $q$ is observed to be comparatively steep. The plot for $\tau(q)$ versus $q$ show marked deviation from the linearity, asserting the presence of the multifractality in the time series for the chosen height. The multifractal measures computed, $A = 1.34$ and $\Delta\alpha = 0.27$, and listed in Table 2. Lastly, the multifractal spectrum is fitted with the p-model (shown with a continuous line). We have obtained a dissipative factor of 0.012, with $p_1 = 0.423$. The p-model fit parameters are listed in Table 2.

It is seen from the above discussion that the multifractal spectrum is sufficient to assess the multifractal nature, henceforth we show the time series and the corresponding multifractal spectrum for the remaining chosen heights. Figure 3 shows the time series selected from the first experiment in the left panel and the corresponding multifractal spectrum in the right panel.

– For the time series corresponding to the mean height of 264.58 km, the multifractal spectrum is slightly right skewed, which can be inferred from measure $A = 0.82$. It indicates the influence of negative moments, $q$, which characterizes smaller fluctuations than the average. Degree of multifractality, $\Delta\alpha = 0.53$. The optimal p-model fit is obtained with parameters $p_1 = 0.364$ and $dp = 0.059$.

– For the time series corresponding to the mean height of 270.22 km, the multifractal spectrum is slightly left skewed, which can be inferred from measure $A = 1.11$. It indicates the influence of positive moments, $q$, which characterize intense larger fluctuations than the average. Degree of multifractality, $\Delta\alpha = 0.82$. The optimal p-model fit is obtained with parameters $p_1 = 0.34$ and $dp = 0.065$.

– For the time series corresponding to the mean height of 292.37 km, the multifractal spectrum is left skewed, reflected in measure $A = 2.99$. It indicates the influence of positive moments, $q$, which characterize intense larger fluctuations than the average. Degree of multifractality, $\Delta\alpha = 0.93$. The optimal p-model fit obtained with parameters $p_1 = 0.339$ and $dp = 0.02$. We could fit the spectrum corresponding to positive values of $q$.

– For the time series corresponding to the mean height of 358.56 km, the multifractal spectrum is right skewed, reflected in measure $A = 0.37$. It indicates the influence of negative moments, $q$, which characterize smaller fluctuations than the average. Degree of multifractality, $\Delta\alpha = 0.52$. The optimal p-model fit obtained with parameters $p_1 = 0.36$ and $dp = 0.07$.

– For the time series corresponding to the mean height of $429.65$ km, the multifractal spectrum is right skewed, also reflected in measure $A = 0.51$. It indicates the influence of negative moments, $q$, which characterize slower varying fluctuations than the average. Degree of multifractality is $\Delta\alpha = 0.28$. The optimal p-model fit obtained with parameters $p_1 = 0.399$ and $dp = 0.0355$.

Figure 4 shows the time series selected from the second experiment in the left panel and the corresponding multifractal spectrum in the right panel.

– For the time series corresponding to the mean height of $348.99$ km, the multifractal spectrum is left skewed, reflected in measure $A = 1.72$. It indicates the influence of positive moments, $q$, which characterize intense larger fluctuations than the average. Degree of multifractality, $\Delta\alpha = 0.22$. The optimal p-model fit obtained with parameters $p_1 = 0.43$ and $dp = 0.006$.

– For the time series corresponding to the mean height of $400.24$ km, the multifractal spectrum is almost symmetrical. This is reflected in measure $A = 0.94$ which is very close to $1$. It indicates both negative and positive moments $q$ characterize larger and smaller fluctuations than the average almost equally. Degree of multifractality, $\Delta\alpha = 0.19$. The optimal p-model fit obtained with parameters $p_1 = 0.4335$ and $dp = 0.01$.

Figure 5 shows a variation of mean density and multifractal width, $\Delta\alpha$ with mean heights for the selected six time series on a 3-dimensional plane. The presence of a plasma bubble characterized by large scale irregularities, which in turn is reflected in the low density, is observed around a mean height of $292.37$ km. Contrarily, stronger multifractality is observed at this height. This inverse variation is in agreement with the turbulent like multiplicative cascade process. On the other hand, as the rocket traversed higher altitudes, the mean density increased while the multifractality became weaker. This suggests that the cascading process resulted in smaller scale irregularities by dissipating energy. Two dimensional plots showing the variation of mean density and $\Delta\alpha$ with mean heights are shown in the inset of Figure 5.

## 5 Concluding remarks

In this work, we investigate the *in situ* F region electric field and electron density measurements obtained from the two experiments carried near the equatorial sites in Brazil using the MFDFA to understand the complexity in the data and to identify the signature of multiplicative energy cascade in irregularities.

In all the time series, we obtained $0.9 < h(q) < 1.5$, which indicates a long range correlation with persistent temporal fluctuations. In addition, we note that the $h(q)$ profile monotonically decreases with respect to $q$, and that $\tau(q)$ shows deviation from the linearity indicating the presence of the multifractality in all time series. Measures of multifractal spectra, $A$ has shown the presence of structures (both smaller and larger) in the fluctuations; and $\Delta\alpha$ has shown weaker to stronger multifractality. The multifractal spectra are fitted with the p-model and we found weight parameter $p_1$ to be different from $0.5$ which confirms the multifractality present in the data. Accounting nonzero dissipation factor suggests that the energy distribution across the

eddies to be nonuniform. Our results show nonhomogenous and intermittent nature of ionospheric irregularities are consistent with previous findings.

In the second experiment, we considered total six time series. Out of which, three time series exhibited monofractal nature and the remaining three showed weaker multifractality which are presented here. $\Delta\alpha$ and skewness are found to be smaller compared to the first experiment. Result for a mean height of 348.99 km is different than the other two heights and evident for some different kinds of physical mechanism which can be described by the multiplicative cascade process. Though time series are characterized by weaker multifractality, this data has fractal behavior with long range correlation. However, we argue that more detailed study is required to reach any definite conclusion on the turbulent-like mechanism driving the ionospheric irregular structures.

Finally, we intend to test the potential of this algorithm in deciphering the morphology of the cascading phenomena. For this, we choose the first experiment where the rocket intercepted a plasma bubble. Muralikrishna et al. (2003) reported the presence of predominant sharp peaks in the power spectra over a wide range of heights, and they attribute these to a developing plasma bubble that subsequently dissipated energy, reaching an equilibrium which is evidenced by the absence of peaks. Our multifractal analysis has captured this sequence of events. The presence of a plasma bubble characterized by large scale irregularities, which in turn is reflected in the low density, is observed around a mean height of 292.37 km. Contrarily, stronger multifractality is observed at this height. This inverse variation is in agreement with the turbulent like multiplicative cascade process. On the other hand, as the rocket traversed higher altitudes, the mean density increased while the multifractality became weaker. This suggests that the cascading process resulted in smaller scale irregularities by dissipating energy.

We conclude at this point where we have presented the schematic hypothesis based on the multifractal analysis of plasma irregularities in the ionospheric F region.

*Acknowledgements.* Authors are grateful to Dr. A. L. Chian for his constructive comments and review on PSD based ionospheric studies which has been adopted in the article. We also thank another anonymous reviewer for his/her comments that has increased the clarity of the article. Finally, NJ is grateful to Dr. Anna Wawrzaszek for the discussion and guidance obtained on the multifractal analysis and p-model. NJ acknowledges the financial assistance received from CAPES. Data used in this analysis is available in the following repository: http://urlib.net/rep/8JMKD3MGP3W34R/3U8PQA8

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

**Table 1.** Multifractal analysis measures for the first experiment : For the time series at mean heights listed in the first column, the second column shows the degree of multifractality ($\Delta\alpha$), the third column gives measure of asymmetry ($A$). Columns 4 to 6 lists the p-model fit parameters, $l_1, p_1, dp$ respectively.

| $<height>$ | degree of multifractality | measure of asymmetry | p-model fit parameters | | |
|---|---|---|---|---|---|
| (km) | $\Delta\alpha$ | $A$ | $l_1$ | $p_1$ | $dp$ |
| 264.58 | 0.53 | 0.82 | 0.5 | 0.364 | 0.059 |
| 270.22 | 0.82 | 1.11 | 0.5 | 0.340 | 0.065 |
| 292.37 | 0.93 | 2.99 | 0.5 | 0.339 | 0.02 |
| 324.00 | 0.72 | 0.32 | 0.5 | 0.315 | 0.090 |
| 358.56 | 0.52 | 0.37 | 0.5 | 0.360 | 0.070 |
| 429.65 | 0.28 | 0.51 | 0.5 | 0.399 | 0.0355 |

**Table 2.** Multifractal analysis measures for the second experiment : For the time series at mean heights listed in the first column, the second column shows degree of multifractality ($\Delta\alpha$), the third column gives measure of asymmetry ($A$). Columns 4 to 6 lists the p-model fit parameters, $l_1, p_1, dp$ respectively.

| $<height>$ | degree of multifractality | measure of asymmetry | p-model fit parameters | | |
|---|---|---|---|---|---|
| (km) | $\Delta\alpha$ | $A$ | $l_1$ | $p_1$ | $dp$ |
| 339.94 | 0.27 | 1.34 | 0.5 | 0.4230 | 0.012 |
| 348.99 | 0.22 | 1.72 | 0.5 | 0.4300 | 0.006 |
| 400.24 | 0.19 | 0.94 | 0.5 | 0.4335 | 0.01 |

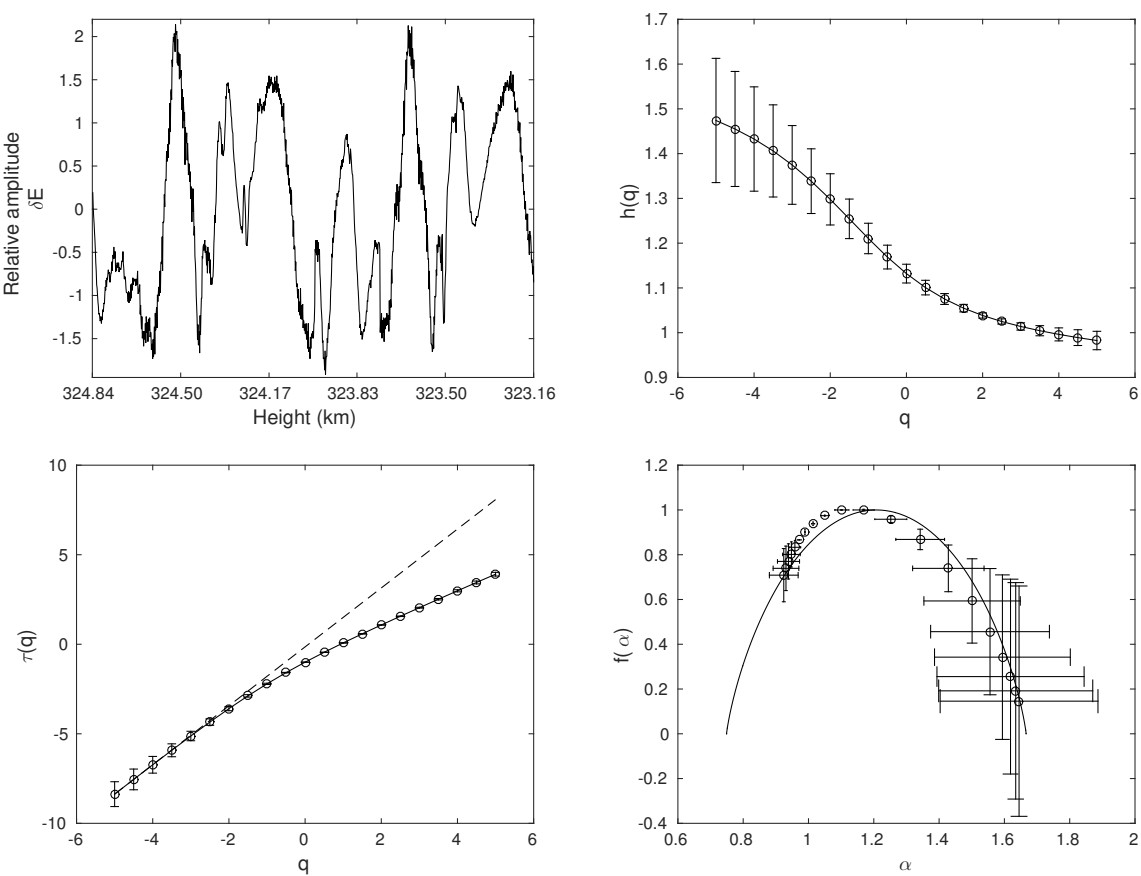

**Figure 1.** Comprehensive MFDFA for the first experiment: upper panel, left figure shows the time series at mean height $324.00$ km and right figure shows $h(q)$ vs $q$ profile. Lower panel, left figure shows $\tau(q)$ vs $q$ profile along with dashed line which represents a linear relationship between $\tau(q)$ and $q$, and right plot is of the multifractal spectrum fitted with the p-model (continuous line).

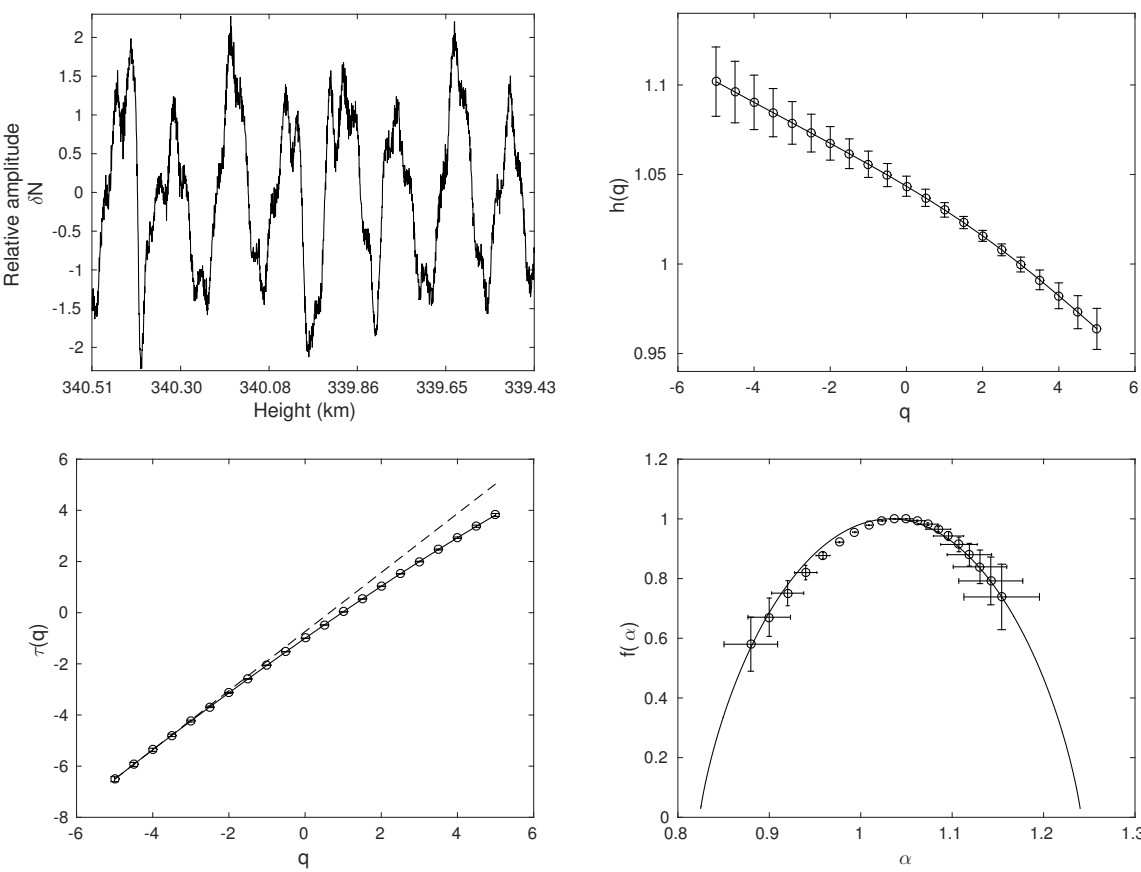

**Figure 2.** Comprehensive MFDFA for the second experiment: upper panel, left figure shows the time series at mean height 339.94 km and right figure shows $h(q)$ vs $q$ profile. Lower panel, left figure shows $\tau(q)$ vs $q$ profile along with dashed line which represents a linear relationship between $\tau(q)$ and $q$, and right plot is of the multifractal spectrum fitted with the p-model (continuous line).

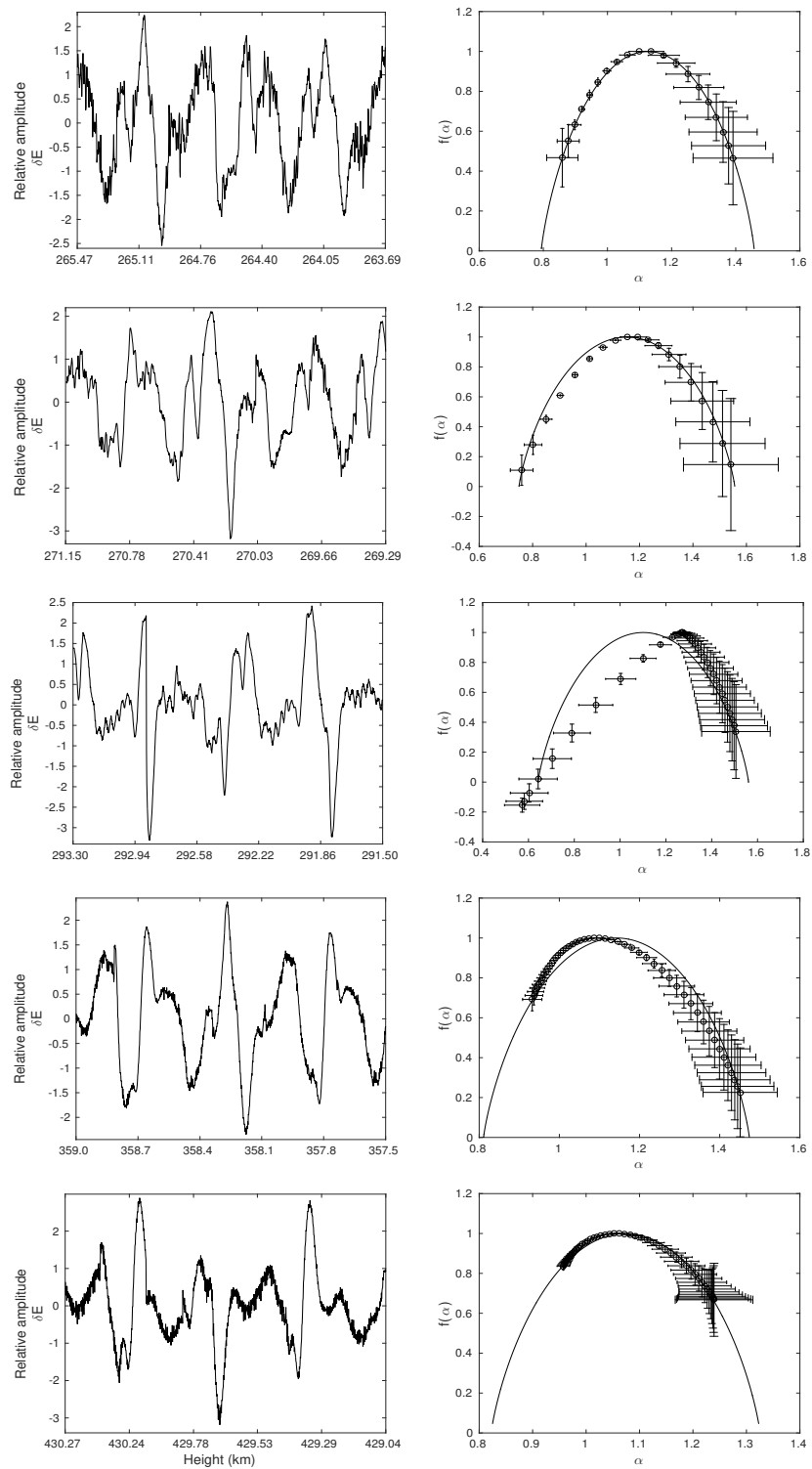

**Figure 3.** MFDFA for the first experiment: top to bottom panel shows the time series and its corresponding multifractal spectrum with the p-model fit (continuous line) for the mean heights of $264.58, 270.22, 292.37, 358.56$ and $429.65$ km respectively.

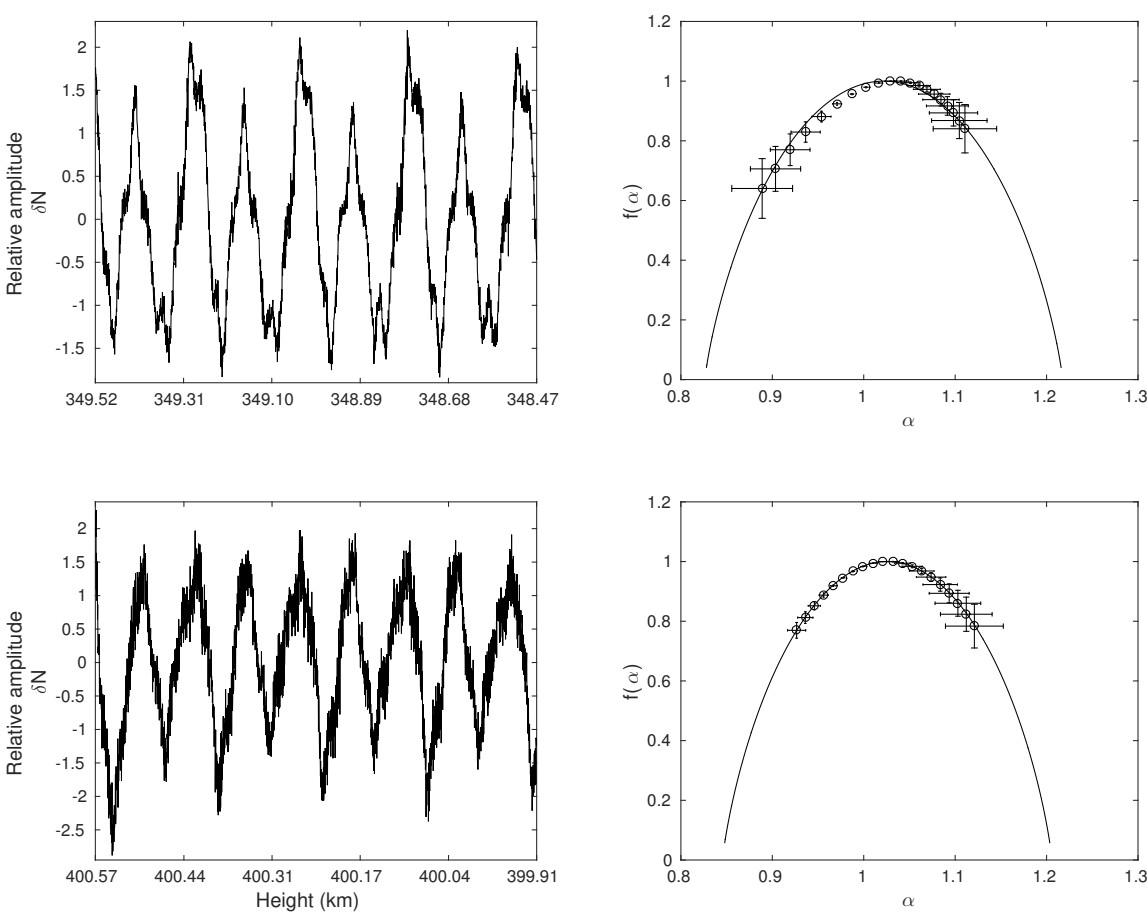

**Figure 4.** MFDFA for the second experiment: Upper panel shows the multifractal analysis of the time series at mean height $348.99$ km and lower panel shows the multifractal analysis of the time series at mean height $400.24$ km. In each panel the left plot shows the time series for given mean height and right plot is of the multifractal spectrum fitted with the p-model (continuous line).

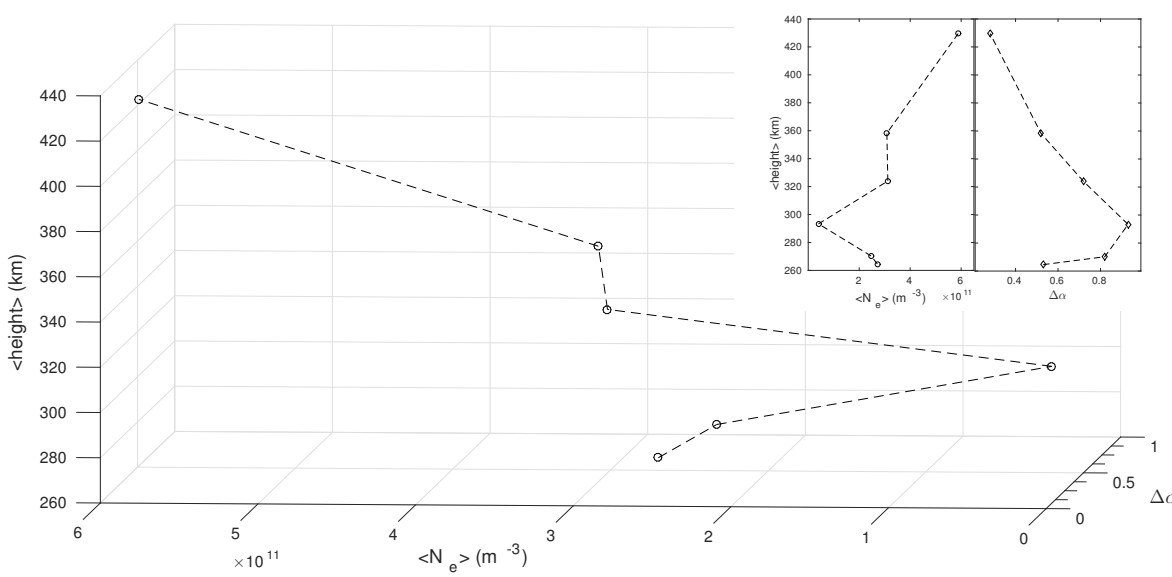

**Figure 5.** Variation of the mean density and the degree of multifractality with the mean height for the selected six time series from the first experiment in a 3-d plane. In the inset, these variations are shown in a 2-d plane: of the mean density (left) and the degree of multifractality (right).