# Peer review of "Structural characterization of the equatorial F region plasma irregularities in the multifractal context"

_Annales Geophysicae, 2019_

## Referee Comment (RC1) · Abraham C.L. Chian (Referee) · 22 Nov 2019

Referee: Abraham Chian, University of Adelaide & Nagoya University

1. General comments

This paper carried out a multifractal study of ionospheric turbulence associated with the equatorial F-region density irregularities. The multifractal detrended fluctuation technique using the p-model fit was applied to analyze several time series of in situ plasma density fluctuations measured by the electric field double-probe onboard two sounding rocket experiments in Alcântara, Brazil. The results of this paper confirmed the nonhomogeneous nature of ionospheric density irregularities and characterized the underlying multiplicative turbulent cascade hypothesis. This paper makes an important contribution to improve our understanding of the multifractal scaling and complex behaviour of the nonlinear evolution of plasma bubbles driven by the generalized Rayleigh-Taylor instability in the equatorial ionosphere. It can be published as a regular article in AG after making the revisions recommended.

2. Specific comments A number of sounding rocket experiments have been launched in Brazil and other sites to study the equatorial plasma bubbles and polar ionosphere apart from the two sounding rocket experiments described by this paper which were not mentioned in this paper. Costa and Kelley (JGR 83(A9) 4359–4364 (1978)) showed that the Rayleigh-Taylor instability that initiates in the bottomside equatorial F-region can nonlinearly develop very sharp gradients leading to the formation of steepened structures responsible for the power-law spectra observed by a rocket experiment in Natal, Brazil. Shock waves were observed by numerical simulation performed by Zargham and Seyler (JGR 92(A9), 10073–10087 (1987)) of the generalized Rayleigh-Taylor instability at the bottomside and topside F-region equatorial ionosphere, which was confirmed by rocket and satellite in situ data reported by Kelley, Seyler and Zargham (JGR 92(A9), 10089–10094 (1987)). Hysell et al. (JGR 99(A5), 8827–8840 (1994a); JGR 99(A5), 8841–8850 (1994b)) proposed a model of plasma steepening, evolving from plasma advection that occurs on the vertical leading edges of plasma depletion wedges, to interpret shock waves detected in the equatorial ionosphere by rockets launched from Kwajalein Atoll. Jahn and Labelle (JGR 103(A10), 23427–23441 (1998)) measured shocklike structures characterized by the density waveforms at the bottomside and topside F-region of the equatorial ionosphere in a rocket experiment in Alcântara, Brazil. To help the readers to understand better the results of this paper, a detailed discussion of the aforementioned papers and the relation between the results of this paper and the previous rocket experiments should be inserted. In addition, a recent paper by Spicher et al. (JGR 120, 10,959-10,978 (2015)) reported a multifractal study of intermittent turbulence in the polar ionosphere

based on a sounding rocket experiment. Since the paper by Spicher et al. (2015) is very closely related to the approach and subject matter of this paper, it is important to insert a discussion to compare the two studies.

Please also note the supplement to this comment:
https://www.ann-geophys-discuss.net/angeo-2019-133/angeo-2019-133-RC1-supplement.pdf

---

## Referee Comment (RC2) · Anonymous Referee #2 · 25 Nov 2019

General comments:

In this paper a multifractal analysis of in situ data of ionospheric irregularities obtained from two experiments is performed. The analysis includes a comparison with the theoretical p-model and confirms the presence of inhomogeneities in the ionospheric plasma. It is also claimed that their results characterize the multiplicative energy cascade in the ionosphere.

Overall, the analysis presented is interesting and the results seem consistent. However, I think that the physical interpretation of the results can be improved, and the grammar needs to be carefully checked. For these reasons I think that this paper can

be published after a major revision is done. A list of general and specific comments, and suggested corrections follows.

1) The authors claim that their results are related to the energy cascade of the turbulent ionospheric plasma. However the paper does not show the presence of an energy cascade in the data. I suggest to include a new figure showing the power spectral density of the data, and indicating the range of scales in which a power-law is observed. In hydrodynamic and solar wind plasma turbulence the power-law has a spectral index around -5/3 $\sim$ -1.67, however it is known that the spectral index of ionospheric turbulence can deviate from this value.

2) The results are interpreted in terms of small-scale and large-scale fluctuations with respect to the average. Do you mean the average value of the data, or the average of the fluctuations? How are these fluctuations and their scale related to plasma fluctuations arising from the generalized Rayleigh-Taylor instability?

3) The grammar needs to be improved. For instance, there are missing articles such as "the" and "a" throughout the text. I indicate some specific examples in the technical corrections below. It is not an exhaustive list though, so please revise the grammar of your manuscript carefully.

Specific comments:

4) The figures should appear on the manuscript in sequential order. For example, on page 6, the analysis jumps from Fig. 1 to Fig. 3.

5) Check the format of citations in the manuscript. For example, in page 3, line 16, "Muralikrishna P. and Abdu M. A. (2006) should be "Muralikrishna and Abdu (2006)".

6) Page 3, line 18: This paragraph should be rewritten to clarify that the key results are from the literature review and not from the present study.

7) Page 4, equation (1): Please define "y" and "y_n".

8) Page 4, equation (2): Why F_q(s) does not depend on "n"? Why the k index does not appear in the summation?

9) Page 4, equation (4): What is h prime?

10) Page 4, line 24: the "&" symbol in "p1&p2" and "l1&l2" represents a special notation? If not, replace with "and". If possible, use "p_1", "p_2", "l_1" and "l_2" to improve the text.

11) Page 5, equation (6): Please explain the "m" and "n" parameters, and rewrite the symbols for the natural logarithms, for example, "lnp1" by "\ln(p_1)".

12) Page 5: line 28: The description of the singularity spectra in terms of a truncated shape, or in terms of left-skewness or right-skewness as done in page 6, needs to be improved. A detailed description will aid potential readers to interpret and understand your results. For example, does it mean that the points accumulate near a certain \alpha value? Why a spectrum truncated to the left indicates insensitivity to large local fluctuations?

13) Page 6, line 15: How is it evident? Please detail.

14) Page 7, line 30: I think that the purpose of this paragraph is to indicate that fractal formalisms can bring about new information wirh respect to classical tools such as the power spectral density. This is a fact that should be stated in the introduction, instead of the conclusions section. Therefore, I suggest the authors to move this paragraph to the introduction. Before that, please clarify what do you mean with "conclusively substantiated the occurrence of [the] energy cascade process in turbulent sites".

15) Page 8, line 3: The following references have also applied fractal and multifractal techniques to characterize the turbulence in the solar and the interplanetary medium. Please consider including them in your literature review.

Abramenko, V. I., Yurchyshyn, V. B., Wang, H., Spirock, T. J. and Goode, P. R. Scaling behavior of structure functions of the longitudinal magnetic field in active regions of the

Sun. Astrophys. J. 577, 487, 2002.

Carbone, V., Bruno, R., Veltri, P. Scaling laws in the solar wind turbulence. Lecture Notes in Physics 462, 153–158, 1995.

Grauer, R., Krug, J., Marliani, C. Scaling of high-order structure functions in magneto-hydrodynamic turbulence. Phys. Lett. A 195, 335–338, 1994.

Chian, A. C.-L., Mu\~noz, P. R. Detection of current sheets and magnetic reconnec-tions at the turbulent leading edge of an interplanetary coronal mass ejection. Astro-phys. J. 733, L34, 2011.

Miranda, R. A., Chian, A. C.-L., Rempel, E. L. Universal scaling laws for fully-developed magnetic field turbulence near and far upstream of the Earth's bow shock. Adv. Space Res. 51, 1893, 2013.

16) The description of the results in the conclusions section needs to be improved. In particular, the paragraph starting on line 15 of page 8 is difficult to understand. What is a "left skewed with right truncated" spectrum?

17) Page 8, line 26: the final paragraph of the conclusions section presents a result which is not mentioned in section 4. Please move the description of Fig. 5 to section 4, and leave the interpretation here.

Technical corrections:

18) In order to improve the presentation of the paper I can suggest the following list of text corrections. Please re-check the grammar carefully.

Page 1, line 3: insert "a" before "multifractal".

Page 1, line 5: insert "The" before "first experiment", and "the" before "second experi-ment".

Page 1, line 6: insert "The" before "multifractal" and before "p-model", and replace "is"

by "are".

Page 1, line 7: insert "The" before "result".

Page 1, line 8: insert "the" before "first experiment".

Page 2, line 10: insert "and" before "multispectral".

Page 2, line 17: replace the "-" with "such as".

Page 2, line 18: remove the "-".

Page 2, line 24: replace "emphin situ" by "\emph{in situ}".

Page 2, line 25: insert "In the" before "first experiment".

Page 2, line 25: remove "is chosen as".

Page 2, line 27: replace ". Whereas" by ", whereas in"

Page 2, line 27: remove the sentence "is chosen as during the experiment".

Page 3, line 6: insert "The" before "SONDA".

Page 3, line 9: insert "A" before "rocket".

Page 3, line 13: insert "The" before "rocket".

Page 3, line 25: rewrite the sentence "Ground equipment, digisonde, near launching station...".

Page 3, line 13: move the sentence "where \alpha represents..." after Eq. (4).

Page 5, line 14: replace "by (Ihlen E. (2012), Table 2)" by "by Table 2 of Ihlen (2012)".

Page 6, line 2: move the sentence "represent the deviation..." after Eq. (8).

Page 8, line 1: replace "model" with "modelling".

Page 8, line 5: remove "past".

Page 8, line 7: remove "altogether".

Page 8, line 8: please rewrite the sentence "supposedly in or near the irregularities".

Page 8: line 8: replace "1.5 > h(q) > 0.9" by "0.9 < h(q) < 1.5". It is easier to read.

Page 8, line 18: remove "(considered in this study)".

Page 8, line 33: replace "in line" with "in agreement".
* * *

---

## Author Comment (AC1) · 4 Jan 2020

Thank you for the feedback and suggestions with detailed references. It has certainly improved the article.

**General Comments:**

1. The authors claim that their results are related to the energy cascade of the turbulent ionospheric plasma. However the paper does not show the presence of an energy cascade in the data. I suggest to include a new figure showing the

power spectral density of the data, and indicating the range of scales in which a power-law is observed. In hydrodynamic and solar wind plasma turbulence the power-law has a spectral index around $-5/3 \sim -1.67$, however it is known that the spectral index of ionospheric turbulence can deviate from this value.

Response: Our aim is to study the nature of the ionospheric irregularities exploring its multifractal scaling. Multifractal scaling is different from the scale sizes obtained through the power spectral density (PSD) analysis. The multifractal analysis identifies the signature of multiplicative energy cascade in the irregularities and the p-model quantifies the scaling measures in the energy cascading process. With the MFDFA and p-model, we are inferring the nonhomogeneous, intermittent nature of time series and how are the fluctuations scaled. The multifractal scaling measures provide information about the dynamics of the data and with these information one can model the time series. Seuront et al. (Journal of Plankton Research, 21(5), 877-922 (1999)) has explained elaborately the multifractal analysis and difference with the power spectral analysis and I quote some of the relevant text from it below: "spectral analysis corresponds to an analysis of variance in which the total variance of a given process is partitioned into contributions arising from processes with different length scales or time scales in the case of spatially or temporally recorded data, respectively. A power spectrum separates and measures the amount of variability occurring in different wavenumber or frequency bands. ... $\beta$ exponent characterizing spectral scale invariance: for instance $\beta = 5/3$ in homogeneous turbulence. The absence of characteristic time scales and the presence of a scaling regime indicate that a multifractal analysis may prove to be successful." As per the suggestion, we have performed the power spectral density (PSD) analysis of all downleg time series. The corresponding results are summarized in the table and placed at end. From the first rocket experiment, time series $< 292.37 >$ km has maximum $\Delta\alpha$ and spectral index for lower frequency $18 - 92$ Hz is observed as $-3.31$. The time series $< 429.65 >$ km has minimum $\Delta\alpha$ and spectral index for lower fre-
quency range $12 - 57$ Hz is observed as $-5.4$. Figures for these two series are placed at end. Previously published papers (Muralikrishna et al., J. Atmos. Sol-Terr. Phy., 65, 1315-1327 (2003)), Muralikrishna and Abdu, Advances in Space Research, 37,1091-1096 (2006)), Muralikrishna and Vieira, Rev. Bras. Geof., 25 (2007)) already analysed the data from the first experiment using the power spectral analysis and showed that the main instability mechanism is the generalized Rayleigh-Taylor instability. We consider these published results and further extend this work using the multifractal analysis. We would like to bring to readers' attention, the important and well accepted fact that ionospheric irregularities do deviate from the Kolmogorov's spectral index $-5/3$ , and thus we are addressing the non-homogeneous nature of ionospheric irregularities. Knowing limitations of the PSD analysis, several papers discussed the advantages of using other methods over PSD and also, to confirm our results with other higher order statistics, we performed the structure function analysis on the time series which has maximum and minimum multifractal width. The structure function analysis shows that both time series deviate from the Kolmogorov scaling $(m/3)$ and also fall below the $m/3$ line, indicating an intermittent behavior (Spicher et al., JGR 120, 10959-10978 (2015)). Time series $< 292.37 >$ km shows the most intermittent behavior. Corresponding figure is placed at end. These complementary analysis and our results are consistent.

2. The results are interpreted in terms of small-scale and large-scale fluctuations with respect to the average. Do you mean the average value of the data, or the average of the fluctuations? How are these fluctuations and their scale related to plasma fluctuations arising from the generalized Rayleigh-Taylor instability?

Response: Authors mean the average of fluctuations. Multifractal spectrum illustrates how segments with small and large fluctuations deviate from the average fractal structure. In multifractal analysis, scales are related to fluctuations in the data. Previously published papers (Muralikrishna et al., J. Atmos. Sol-

[Figure]

Terr. Phy., 65, 1315-1327 (2003)), Muralikrishna and Abdu, Advances in Space Research, 37,1091-1096 (2006)), Muralikrishna and Vieira, Rev. Bras. Geof., 25 (2007)) already analysed the data from the first experiment using the power spectral analysis and showed that main instability mechanism is the generalized Rayleigh-Taylor instability. We consider this published results and further extended this work by the multifractal analysis. We would like to take readers' attention to the important and well accepted fact that ionospheric irregularities do deviate from the Kolmogorov's spectral index $-5/3$ , and thus addressing the non-homogeneous nature of ionospheric irregularities and also, these irregularities are found to be intermittent.

3. The grammar needs to be improved. For instance, there are missing articles such as "the" and "a" throughout the text. I indicate some specific examples in the technical corrections below. It is not an exhaustive list though, so please revise the grammar of your manuscript carefully.

Response: Manuscript has been revised for correct grammar.

**Specific comments:**

4. The figures should appear on the manuscript in sequential order. For example, on page 6, the analysis jumps from Fig. 1 to Fig. 3.

Response: In the manuscript, now figures are ordered in sequence.

5. Check the format of citations in the manuscript. For example, in page 3, line 16, "Muralikrishna P. and Abdu M. A. (2006) should be "Muralikrishna and Abdu (2006)".

Response: Citation format has been revised for all the references.

6. Page 3, line 18: This paragraph should be rewritten to clarify that the key results are from the literature review and not from the present study.

Response: yes, "aforementioned analysis" word has been added for clarity. Also references are cited just after this word instead of keeping at the end of the paragraph (page 4, line 14).

7. Page 4, equation (1): Please define "$y$" and "$y_n$".

Response: we have omitted few steps while describing the MFDFA method. Now the MFDFA method has been improved and have added all missing equations.

8. Page 4, equation (2): Why $F_q(s)$ does not depend on "n"? Why the k index does not appear in the summation?

Response: We have omitted few steps while describing the MFDFA method. Now the MFDFA method has been improved and have added all missing equations.

9. Page 4, equation (4): What is h prime?

Response: h prime means first derivative of h with respect to q.

10. Page 4, line 24: the "$\&$" symbol in "$p1\&p2$" and "$l1\&l2$" represents a special notation? If not, replace with "and". If possible, use "$p_1$", "$p_2$", "$l_1$" and "$l_2$" to improve the text.

Response: $\&$ is replaced by "and", $p1$ is replaced by $p_1$, $p2$ is replaced by $p_2$, $l1$ is replaced by $l_1$, and $l2$ is replaced by $l_2$.

11. Page 5, equation (6): Please explain the "m" and "n" parameters, and rewrite the symbols for the natural logarithms, for example, $lnp1$ by $ln(p_1)$.

Response: $lnp1$ has been replaced by $ln(p_1)$ in equation 6. In the equation "$n''$ is number of generations in the binary fragmentation process. In each generation,
$l_1^m l_2^{(n-m)}$ gives size of a segment, where, notation $m$ is number of left side fragment and $n-m$ represents right side fragment in a segment (Halsey et al., Phys. Rev. A, 33,1141-1151 (1986)). This is now included in the manuscript (line 5 on Page 6).

12. Page 5: line 28: The description of the singularity spectra in terms of a truncated shape, or in terms of left-skewness or right-skewness as done in page 6, needs to be improved. A detailed description will aid potential readers to interpret and understand your results. For example, does it mean that the points accumulate near a certain $\alpha$ value? Why a spectrum truncated to the left indicates insensitivity to large local fluctuations?

Response: A detailed description has been added in the manuscript (page 7, line 1 to page 7, line 25) To avoid confusion, we omit "truncation" and retain "skewness".

Please see the text below:

In the MFDFA, fluctuation function $Fq(s)$ is obtained by computing $q^{th}$ order local root mean square (RMS) for multiple segment size, i.e., for scales $s$. A segment may contain smaller to larger fluctuations. Rapid variation in fluctuations influence overall RMS for smaller scale sizes whereas slow variation in fluctuations influence overall RMS for larger scale sizes. Negative $q$ values characterize smaller fluctuations and positive $q$ values characterize larger fluctuations in a segment. When $q = 0$, it behaves neutral. $h(q)$ has dependence on $q$. To outline, for a multifractal time series $h(q)$ monotonically decreases with $q$, and $\tau(q)$ shows nonlinear dependence on $q$. With $q = 0$ as a center point, let us inspect how $h(q)$ varies with respect to negative and positive values of $q$. If time series is influenced by smaller fluctuations, then variation of $h(q)$ for negative $q$ will be faster, i.e., a steeper slope can be observed with respect to negative $q$ and vice-versa (Kantelhardt2002, Ihlen2012). The multifractal spectrum illustrates how segments with small and large fluctuations deviate from the average fractal structure. The shape

and width of multifractal spectrum are also important measures to quantify the nature of multifractality present in the data. For $f(\alpha) = 1$, the corresponding value of $\alpha$, known as $\alpha_0$ divides the spectrum into left and right sides. A shape of the spectrum, difference in left and right side of the spectrum, can be quantified by measure of asymmetry, $A$, is given by

$$A = \frac{\alpha_0 - \alpha_{min}}{\alpha_{max} - \alpha_0} \tag{1}$$

When $A = 1$, the multifractal spectrum is symmetric in the sense that the time series is influenced by both larger as well as smaller fluctuations. When $A > 1$, the spectrum is left-skewed which implies that the time series is more influenced by the larger fluctuations. When $A < 1$, the spectrum is right-skewed which implies that the time series is more influenced with smaller fluctuations. A width of the spectrum can be quantified by $\Delta\alpha$, which is the difference between maximum and minimum dimension.

$$\Delta\alpha = \alpha_{max} - \alpha_{min} \tag{2}$$

The width of the spectrum infers the degree of multifractality and complexity of the data. It represents the deviation from the average fractal structure, and directly relates to the parameters corresponding to the multiplicative cascade process. Larger (smaller) value of $\Delta\alpha$ infers stronger (weaker) multifractality in the data. The multifractal spectrum reflects the characteristics of the $h(q)$ profile. In the spectrum, contrary to the $h(q)$ profile, left side is characterized by positive values of $q$, and right side is characterized by negative values of $q$. When the $h(q)$ profile show steeper variations on left side, i.e., for negative $q$'s, right side of the spectrum shows faster variation compared to its left side.

13. Page 6, line 15: How is it evident? Please detail.

Response: In the submitted version, we did describe a relation between observation in the following line. Now, the edited interpretation in response to Q12 will make clear the relation between h(q) and multifractal spectrum.

14. Page 7, line 30: I think that the purpose of this paragraph is to indicate that fractal formalisms can bring about new information wirh respect to classical tools such as the power spectral density. This is a fact that should be stated in the introduction, instead of the conclusions section. Therefore, I suggest the authors to move this paragraph to the introduction. Before that, please clarify what do you mean with "conclusively substantiated the occurrence of [the] energy cascade process in turbulent sites".

Response: We have edited the indicated text in the Conclusion section by inserting it in the Introduction section but with care of not repeating any point previously stated. For the phrase "conclusively substantiated the occurrence of [the] energy cascade process in turbulent sites", the authors intend to emphasize that the theory is tested against the observations and controlled laboratory experiments, and found to be in good agreement.

15. Page 8, line 3: The following references have also applied fractal and multifractal techniques to characterize the turbulence in the solar and the interplanetary medium. Please consider including them in your literature review.

Response: Suggested references have been added in the manuscript in the "Introduction" section. Page 2, line 19.

16. The description of the results in the conclusions section needs to be improved. In particular, the paragraph starting on line 15 of page 8 is difficult to understand. What is a "left skewed with right truncated" spectrum?

Response: We have improved this paragraph. page 10, lines 2-9.

17. Page 8, line 26: the final paragraph of the conclusions section presents a result which is not mentioned in section 4. Please move the description of Fig. 5 to section 4, and leave the interpretation here.

Response: The description of Figure 5 is moved to the Result section and interpretation is kept as it is in the Conclusion section.

**Technical corrections:**

18. In order to improve the presentation of the paper I can suggest the following list of text corrections. Please re-check the grammar carefully.

Response: Thank you for the corrections. We have carefully rechecked the grammar.

**Table 1.** Power spectral analysis for the both rocket experiments : For the time series at mean heights listed in the first column, the second column shows the first spectral exponent $\beta_1$, the third column gives corresponding lower frequency range. Columns 4 shows multifractal width ($\Delta\alpha$) obtained in the MFDFA, respectively.

| $< height >$ (km) | $\beta_1$ | lower frequency range (Hz) | $\Delta\alpha$ |
|---|---|---|---|
| Rocket experiment 1 | | | |
| 264.58 | -3.67 | 12-73 | 0.53 |
| 270.22 | -3.58 | 12-73 | 0.82 |
| 292.37 | -3.31 | 18-92 | 0.93 |
| 324.00 | -3.44 | 10-78 | 0.72 |
| 358.56 | -4.08 | 10-78 | 0.52 |
| 429.65 | -5.40 | 12-57 | 0.28 |
| Rocket experiment 2 | | | |
| 339.94 | -3.41 | 10-80 | 0.53 |
| 348.99 | -3.19 | 10-100 | 0.82 |
| 400.24 | -2.91 | 10-80 | 0.93 |

[Figure]

**Figures**:

Fig 1. PSD analysis for downleg time series at mean height $< 292.37 >$ km for which maximum $\Delta\alpha$ is obtained

Fig 2. PSD analysis for downleg time series at mean height $< 429.65 >$ km for which minimum $\Delta\alpha$ is obtained

Fig 3. Structure function analysis of above mentioned series for order m = 1 to 4

[Figure]

**Fig. 1.**

[Figure]

**Fig. 2.**

[Figure]

**Fig. 3.**

---

## Author Comment (AC2) · 4 Jan 2020

Thank you Dr. Abraham Chian for the feedback, suggestions with detailed references and also for the endorsement. It has certainly improved the article.

**Specific Comments:**

A number of sounding rocket experiments have been launched in Brazil and other sites to study the equatorial plasma bubbles and polar ionosphere apart from the two sounding rocket experiments described by this paper which were not mentioned

in this paper. Costa and Kelley (JGR 83(A9) 4359–4364 (1978)) showed that the Rayleigh-Taylor instability that initiates in the bottomside equatorial F-region can non-linearly develop very sharp gradients leading to the formation of steepened structures responsible for the power-law spectra observed by a rocket experiment in Natal, Brazil. Shock waves were observed by numerical simulation performed by Zargham and Seyler (JGR 92(A9), 10073–10087 (1987)) of the generalized RayleighTaylor instability at the bottomside and topside F-region equatorial ionosphere, which was confirmed by rocket and satellite in situ data reported by Kelley, Seyler and Zargham (JGR 92(A9), 10089–10094 (1987)). Hysell et al. (JGR 99(A5), 8827–8840 (1994a); JGR 99(A5), 8841–8850 (1994b)) proposed a model of plasma steepening, evolving from plasma advection that occurs on the vertical leading edges of plasma depletion wedges, to interpret shock waves detected in the equatorial ionosphere by rockets launched from Kwajalein Atoll. Jahn and Labelle (JGR 103(A10), 23427–23441 (1998)) measured shocklike structures characterized by the density waveforms at the bottomside and topside F-region of the equatorial ionosphere in a rocket experiment in Alcântara, Brazil. To help the readers to understand better the results of this paper, a detailed discussion of the aforementioned papers and the relation between the results of this paper and the previous rocket experiments should be inserted. In addition, a recent paper by Spicher et al. (JGR 120, 10,959-10,978 (2015)) reported a multifractal study of intermittent turbulence in the polar ionosphere based on a sounding rocket experiment. Since the paper by Spicher et al. (2015) is very closely related to the approach and subject matter of this paper, it is important to insert a discussion to compare the two studies.

Response:
As per the suggestions provided in the specific comments section, we have included all the references and adopted your discussion on the PSD studies reported earlier.

In addition, we performed power spectral analysis and structure function analysis of

two downleg time series from the first experiment for which maximum and minimum multifractal width, , was observed, to check whether our results are consistent with the previous works. We found our results to be consistent. Time series $< 292.37 >$ km has maximum $\Delta\alpha$ and spectral index $-3.31$ is observed in the frequency range $18 - 92$ Hz. Time series $< 429.65 >$ km has minimum $\Delta\alpha$ and spectral index $-5.4$ is observed in the frequency range $12 - 57$ Hz. Structure function analysis shows that both time series deviate from the Kolmogorov scaling $(m/3)$ and fall below the $m/3$ line, indicating intermittent behavior. Time series $< 292.37 >$ km shows the most intermittent behaviour. These figures are placed at end.

We have added three paragraphs in the Introduction section. Please find the additional paragraphs below:

page 2, line 3:
Various rocket experiments and numerical simulations have been performed and contributed to our understanding of the generation and development of ionospheric irregularities and possible instabilities causing mechanism. Costa and Kelley (JGR 83(A9) 4359–4364 (1978)) showed that the Rayleigh-Taylor instability that initiates in the bottomside equatorial F-region can nonlinearly develop very sharp gradients leading to the formation of steepened structures responsible for the power-law spectra observed by a rocket experiment in Natal, Brazil. Shock waves were observed by numerical simulation performed by Zargham and Seyler (JGR 92(A9), 10073–10087 (1987)) of the generalized RayleighTaylor instability at the bottomside and topside F-region equatorial ionosphere, which was confirmed by rocket and satellite in situ data reported by Kelley, Seyler and Zargham (JGR 92(A9), 10089–10094 (1987)). Hysell et al. (JGR 99(A5), 8827–8840 (1994a); JGR 99(A5), 8841–8850 (1994b)) proposed a model of plasma steepening, evolving from plasma advection that occurs on the vertical leading edges of plasma depletion wedges, to interpret shock waves detected in the equatorial ionosphere by rockets launched from Kwajalein Atoll. Jahn

and Labelle (JGR 103(A10), 23427–23441 (1998)) measured shocklike structures characterized by the density waveforms at the bottomside and topside F-region of the equatorial ionosphere in a rocket experiment in Alcântara, Brazil.

Page 2, line 28 :
Structure function analysis performed on ionospheric in situ data have revealed the intermittent nature of ionospheric irregularities owing to the large deviations from the Kolmogorov's K41 universal power-law index proposed for neutral fluid turbulence (Spicher et al., JGR 120, 10959-10978 (2015)).

page 2, line 31:
In all the above mentioned studies, the main feature that gets highlighted is that the power spectra point to large deviations from the homogeneous turbulence described by the Kolmogorov spectrum ($-5/3$). Also, higher order statistics like structure function analysis confirmed the deviation from the Kolmogorov scales. Thus affirming non-homogeneity and intermittency in ionospheric irregularities. In the complex scenario of ionospheric turbulence, an important question that arises in the context of this paper is, "is non-homogeneity, which can be characterized by multifractal spectra, a cause for the large deviations from $-5/3$?" To answer this question, we propose to use the multifractal detrended fluctuation analysis (MFDFA) on the equatorial F region plasma irregularities.

**Table 1.** Power spectral analysis for the both rocket experiments : For the time series at mean heights listed in the first column, the second column shows the first spectral exponent $\beta_1$, the third column gives corresponding lower frequency range. Columns 4 shows multifractal width ($\Delta\alpha$) obtained in the MFDFA, respectively.

| $< height >$ (km) | $\beta_1$ | lower frequency range (Hz) | $\Delta\alpha$ |
|---|---|---|---|
| Rocket experiment 1 | | | |
| 264.58 | -3.67 | 12-73 | 0.53 |
| 270.22 | -3.58 | 12-73 | 0.82 |
| 292.37 | -3.31 | 18-92 | 0.93 |
| 324.00 | -3.44 | 10-78 | 0.72 |
| 358.56 | -4.08 | 10-78 | 0.52 |
| 429.65 | -5.40 | 12-57 | 0.28 |
| Rocket experiment 2 | | | |
| 339.94 | -3.41 | 10-80 | 0.53 |
| 348.99 | -3.19 | 10-100 | 0.82 |
| 400.24 | -2.91 | 10-80 | 0.93 |

**Figures:**

Fig 1. PSD analysis for downleg time series at mean height $< 292.37 >$ km for which maximum $\Delta\alpha$ is obtained

Fig 2. PSD analysis for downleg time series at mean height $< 429.65 >$ km for which minimum $\Delta\alpha$ is obtained

Fig 3. Structure function analysis of above mentioned series for order m = 1 to 4

———————————————————

**Fig. 1.**

[Figure]

[Figure]

**Fig. 2.**

Fig. 3.

---

## Author Response (AR1)

Point to point responses to both reviewers have been posted already.

Changes made in the manuscript as per suggestions by the first reviewer, Dr. Chian Abraham C.L. has been presented in the blue colour:
1) page 2, lines 2 – 12
2) page 2, line 27 to page 3, line 2

Changes made in the manuscript as per suggestions by the second reviewer has been presented in the red colour.
1) page 4, lines 24 – 26
2) page 4, line 30 to page 5, line 22
3) page 6, lines 5 – 7
4) page 6, line 28 and line 29
5) page 7, lines 1 – 12
6) page 7, line 20
7) page 9, lines 15 – 22
8) page 10, lines 1 – 9
9) other grammatical corrections are kept in red colour.

---

## Referee Report (RR1)

The authors have answered all of my questions and the paper has been greatly improved. Therefore, it can be accepted for publication.

I have a final suggestion which does not compromise the acceptance of this paper. The authors have included a table and three figures in their response in the interactive discussions. Since this information is apparently new and unpublished, I would like to suggest to include them as an supplement to the main article. I leave this for the authors to decide.